# Impact of Sarcopenia on Patients with Localized Pancreatic Ductal Adenocarcinoma Receiving FOLFIRINOX or Gemcitabine as Adjuvant Chemotherapy

**DOI:** 10.3390/cancers14246179

**Published:** 2022-12-14

**Authors:** Victor Mortier, Felix Wei, Anna Pellat, Ugo Marchese, Anthony Dohan, Catherine Brezault, Maxime Barat, David Fuks, Philippe Soyer, Romain Coriat

**Affiliations:** 1Gastroenterology and Digestive Oncology Unit, Cochin Hospital AP-HP, 75014 Paris, France; 2Department of Radiology, Cochin Hospital AP-HP, 75014 Paris, France; 3Institut Cochin, Université de Paris, INSERM U 1016 CNRS UMR 8104, 75014 Paris, France; 4Digestive Surgery Department, Cochin Hospital AP-HP, 75014 Paris, France

**Keywords:** pancreatic adenocarcinoma, adjuvant chemotherapy, FOLFIRINOX, gemcitabine, sarcopenia

## Abstract

**Simple Summary:**

Pancreas cancer will become the second deadliest cancer in 2030. One-third of patients with pancreatic cancer are treated with surgery followed by intravenous chemotherapy. This aggressive treatment has to be delivered to people fit enough to receive it. Many variables are used to define this status, such as performance status, albuminemia or sarcopenia. In our study, we calculated the sarcopenia status by measuring via computer tomography the area of the psoas; if it is low in terms of sex and BMI, the patient is considered sarcopenic. We found out that sarcopenic patients with operated pancreatic cancer have a lower overall survival no matter the type of chemotherapy used.

**Abstract:**

Background: Despite its toxicity, modified FOLFIRINOX is the main chemotherapy for localized, operable pancreatic adenocarcinomas. Sarcopenia is known as a factor in lower overall survival (OS). The purpose of this study was to assess the impact of sarcopenia on OS in patients with localized pancreatic ductal adenocarcinoma (PDAC) who received modified FOLFIRINOX or gemcitabine as adjuvant chemotherapy. Methods: Patients with operated PDAC who received gemcitabine-based (GEM group) or oxaliplatin-based (OXA group) adjuvant chemotherapy between 2008 and 2021 were retrospectively included. Sarcopenia was estimated on a baseline computed tomography (CT) examination using the skeletal muscular index (SMI). The primary evaluation criterion was OS. Secondary evaluation criteria were disease-free survival (DFS) and toxicity. Results: Seventy patients treated with gemcitabine-based (*n* = 49) and oxaliplatin-based (*n* = 21) chemotherapy were included, with a total of fifteen sarcopenic patients (eight in the GEM group and seven in the OXA group). The median OS was shorter in sarcopenic patients (25 months) compared to non-sarcopenic patients (158 months) (*p* = 0.01). A longer OS was observed in GEM non-sarcopenic patients (158 months) compared to OXA sarcopenic patients (14.4 months) (*p* < 0.01). The median OS was 157.7 months in the GEM group vs. 34.1 months in the OXA group (*p* = 0.13). No differences in median DFS were found between the GEM group and OXA group. More toxicity events were observed in the OXA group (50%) than in the GEM group (10%), including vomiting (*p* = 0.02), mucositis (*p* = 0.01) and neuropathy (*p* = 0.01). Conclusion: Sarcopenia is associated with a worse prognosis in patients with localized operated PDAC whatever the delivered adjuvant chemotherapy.

## 1. Introduction

Pancreatic adenocarcinoma is a major cause of cancer-related death with poor overall survival (OS) at five years [1]. In 2030, pancreatic adenocarcinoma is estimated to become the second leading cause of cancer-related death [2]. Approximately one-third of patients with pancreatic adenocarcinomas are resectable upon diagnosis and eligible for surgery, one-third are already metastatic and the other third are locally advanced and potentially resectable after neoadjuvant therapy [3]. 

For many years, adjuvant therapy in patients with resected pancreatic ductal adenocarcinoma (PDAC) was limited to gemcitabine monotherapy. Recently, studies have demonstrated that other combinations of chemotherapy could be used, resulting in improved OS compared to gemcitabine [4,5,6]. In a phase 3 randomized study, Neoptolemos et al. first highlighted that the adjuvant combination of gemcitabine and capecitabine significantly improved the median OS compared to gemcitabine alone (28 months vs. 25.5 months) [4]. Conroy et al. advocated the modified FOLFIRINOX regimen as the new standard of care in the adjuvant setting, considering it led to a significantly longer OS than gemcitabine among patients with resected pancreatic cancer [5]. Indeed, the median OS was 54.4 months in the modified FOLFIRINOX group vs. 35.0 months in the gemcitabine group (*p* = 0.01) [5]. Finally, the association of nab-paclitaxel and gemcitabine yielded a slight benefit in terms of OS, but not disease-free survival (DFS) compared to gemcitabine alone in patients with resected PDAC so modified FOLFIRINOX remains the standard of care in patients fit enough to receive it [6]. 

Low-performance status and sarcopenia have been identified as variables associated with lower OS in patients with cancer [7,8,9]. Sarcopenia, defined as the skeletal muscle mass and strength loss, is due, among aging and other causes, to an increased tumor-related metabolism, an exocrine insufficiency or/and poor oral intake. Studies found that sarcopenia is an indicator of cachexia, which also considers weight loss and a low body mass index (BMI) [10]. Sarcopenia is found in 35–65% of patients with pancreatic cancer, which represents a large proportion of patients for whom the recognition of this status may be useful for future care [7,11,12]. 

The purpose of this study was to evaluate the impact of sarcopenia and adjuvant chemotherapy in localized PDAC treated with a curative intent. 

## 2. Method 

### 2.1. Patients

Patients who underwent surgery for localized PDAC in a tertiary center (Cochin University Hospital) were included. Inclusion criteria were: (i) age > 18 years and (ii) histologically confirmed localized PDAC with no metastasis and no contraindication to surgery. All patients underwent surgery between 1 January 2004 and 31 December 2021. Exclusion criteria were patients without adjuvant therapy, with non-curative surgery and with non-pancreatic lesions. All surgeries were validated during a multidisciplinary meeting which included oncologists, digestive oncologists, pancreatic surgeons and radiologists who confirmed the curative intent. 

Patients were further assigned to the gemcitabine-based chemotherapy (GEM group) and oxaliplatin-based chemotherapy (OXA group) groups. In the GEM group, patients received either gemcitabine alone or gemcitabine plus capecitabine. The gemcitabine regimen corresponded to 1000 mg/m^2^ administered on days 1, 8 and 15 for a 4-week cycle. The gemcitabine–capecitabine regimen corresponded to a gemcitabine regimen of 800 mg/m^2^ and oral capecitabine twice a day every day from day 1 to day 21, for a 4-week cycle. In the OXA group, patients received either FOLFOX or modified FOLFIRINOX regimen. FOLFOX regimen corresponded to 85 mg/m^2^ oxaliplatin, 400 mg/m^2^ leucovorin and 2400 mg/m^2^ fluorouracil administered every 14 days. The modified FOLFIRINOX regimen corresponded to the FOLFOX regimen and 150 mg/m^2^ of irinotecan administered every 14 days [5]. The assigned group was chosen by the assumed capacity of the patient to tolerate the treatment and the date of inclusion, as, before 2018, FOLFIRINOX was not a standard of treatment.

The present study was in accordance with the Declaration of Helsinki and was approved by our local ethics committee (Comité Local d’Ethique des publications de l’hôpital Cochin; CLEP AAA-2022-08020) according to French regulations. All data were collected from medical files and reported in an online Case Report Form.

### 2.2. Body Composition Evaluation

Measurements of muscle surface were performed by a radiologist with five years of experience in abdominal radiology (M.B.) blinded to patient data and outcomes. Measurements of muscle surface included the psoas, erector spinae, quadratus lumborum, transversus abdominis, external and internal oblique and rectus abdominis muscles at the level of the third lumbar vertebra (i.e., L3) on unenhanced computed tomography (CT) images. For all CT examinations, images were obtained with a 64-section CT unit (collimation thickness, 0.625 mm; matrix size, 512 × 512; tube current, 120 kV). Semi-automatic segmentation was used based on attenuation value thresholds with minimal manual adjustments when needed.

Sarcopenia was defined at baseline on the preoperative CT by the skeletal muscular index (SMI) [13]. SMI corresponds to the area of all abdominal muscles on the CT at the level of the third lumbar vertebra divided by square height. For all patients, sarcopenia was defined by a score of less than 41 cm^2^/m^2^, 43 cm^2^/m^2^ and 53 cm^2^/m^2^ in women, men with a body mass index (BMI) less than 25 kg/m^2^ and men with a BMI above 25 kg/m^2^, respectively (Figure 1) [13]. 

The primary objective of the study was OS. Clinical and radiological evaluations were performed every four cycles of treatment during chemotherapy and every three months after chemotherapy until disease recurrence or death. OS was defined as the time from surgery to death. The secondary endpoints were DFS and toxicity. DFS was defined as the time from surgery to the first cancer-related event or death. Patients without recurrence at the time of analysis or loss of follow-up had their data censored on the date of the last informative follow-up. 

### 2.3. Statistical Analysis

Descriptive statistics (median, ranges and interquartile range (IQR)) were used to report patient baseline characteristics and treatment-induced adverse events. Differences between the GEM and OXA groups were evaluated with Fisher’s exact test, the chi-square (χ^2^) test with Yates correction, or Student’s *t* test when appropriate. Survival analyses were performed using the Kaplan–Meier method with the log-rank test. A *p* value < 0.05 was considered to indicate statistically significant differences. Calculations were performed with NCSSC 2007 software (NCSS, Kaysville, UT, USA). 

## 3. Results

### 3.1. Adjuvant Therapy

Between 2008 and 2021, a total of 70 patients received adjuvant chemotherapy after surgery for localized PDAC, 21 in the OXA group and 49 in the GEM group (Figure 2). There were 33 men and 37 women, with a median age of 68 years. The median ages were 67 years (range: 45–85) and 72 years (range: 43–81) in the GEM and OXA groups, respectively (Table 1). Fifteen patients (15/70) met the criteria for sarcopenia, eight and seven in the GEM and OXA group, respectively. In the GEM group, PDACs were pT1, pT2, pT3 and pT4 in 4% (2/49), 24% (12/49), 62% (30/49) and 10% (5/49) of patients, respectively, according to the results of surgical pathology. In the OXA group, pT2 and pT3 were identified in 57% (12/21) and 43% (9/21) patients, respectively. Surgical examination identified an R0 resection in 90% (44/49) and 100% (21/21) of patients in the GEM and OXA groups, respectively. All non-R0 PDACs in the GEM group were R1 (10%; 5/49). No PDACs were R2. The median duration of chemotherapy was significantly longer in the GEM group (6 months) than in the OXA group (4.5 months) (*p* < 0.01), corresponding to six and nine cycles of chemotherapy. Patients’ demographics and disease characteristics are described in Table 1. 

No difference in the median OS was found between the GEM group (157.7 months; IQR: 30 not reached) and the OXA group (34.1 months; IQR: 25–45) (*p* = 0.13). No difference in the OS rate at 24 months was found (82% vs. 71%, respectively) (Figure 3A). There was no difference in median DFS between the GEM group (22.7 months; IQR: 10.3–57.7) and the OXA group (14.4 months; IQR: 7.6–25.5) (*p* = 0.11) (Figure 3B). In the 36 patients with recurrence in the GEM group, 47% (17/36) had an exclusive local recurrence, and 38% (14/36) had only distant metastasis, significantly different from the OXA group with 30% (3/10) and 20% (2/10), respectively. Furthermore, 13% (5/36) of the GEM group had synchronous metastasis and local recurrence vs. 50% (5/10) of the OXA group (*p* = 0.04).

### 3.2. Sarcopenia

A total of 15 patients (26%) were sarcopenic and 43 (74%) were non-sarcopenic. The median age of sarcopenic patients was 73 years vs. 65 years for non-sarcopenic patients (*p* = 0.03) (Table 2).

In the non-sarcopenic group, tumors were classified as pT1, pT2, pT3 and pT4 at the histopathological examination in 5% (2/43), 29% (12/43), 60% (26/43) and 7% (3/43) of patients, respectively. In the sarcopenic group, tumors were classified as pT2 and pT3 in 53% (8/15) and 47% (7/15) of patients, respectively.

Surgical examination identified an R0 resection in 93% (40/43) and 93% (14/15) of patients in the non-sarcopenic group and in the sarcopenic group, respectively. The non-R0 patients were all R1 in the non-sarcopenic group (7%; 3/43) and in the sarcopenic group (7%; 1/15). The median duration of chemotherapy was 6 months in both groups. Patients’ demographics and disease characteristics are described in Table 2.

A significant difference in the median OS was found between sarcopenic (25.5 months) and non-sarcopenic patients (157.7 months) (*p* = 0.01), with a hazard ratio of 6.9 (95% CI: 1.67–28.29). The overall survival rates at 1 year, 2 years and 5 years were retrospectively 91%, 66% and 16% for sarcopenic patients, and 92%, 87% and 64% for non-sarcopenic patients (Figure 4A). No significant difference in median DFS was found between non-sarcopenic (20.9 months; IQR: 7.4–47.6) and sarcopenic patients (14.6 months; IQR: 6.6–25.9) (*p* = 0.34) (Figure 4B).

### 3.3. Sarcopenia and Chemotherapy Group

Among the fifteen sarcopenic patients, seven (7/15; 47%) were in the OXA group and eight (8/15; 53%) were in the GEM group. Among the 43 non-sarcopenic patients, 11 (11/43; 26%) were in the OXA group and 32 (32/43; 74%) were in the GEM group.

The median OS was 157.7 months, 43.4 months, 40.4 months and 14.4 months in the GEM non-sarcopenic group, the OXA non-sarcopenic group, the GEM sarcopenic group and the OXA sarcopenic group (*p* < 0.01), respectively. Kaplan–Meier curves are represented in Figure 5. The median DFS was 26.6 months in the GEM sarcopenic group, 20.9 months in the GEM non-sarcopenic group, 15.6 months in the OXA non-sarcopenic group and 9.3 months in the OXA sarcopenic group (*p* = 0.07) (Figure 5).

### 3.4. Tumor Status

Significant differences in median DFS were found between pN0 (66.8 months), pN1 (43.4 months) and pN2 patients (27.4 months) (*p* = 0.02). Significant differences in median DFS were found between R0 patients (41 months) and R1 patients (11.3 months) (*p* < 0.01). No differences were found in DFS for the other statuses (Table 3).

The median OS was significantly different in R0 patients (158 months) vs. R1 patients (not reached but lower) (*p* = 0.04). The median OS was not different according to tumor and node status (Table 3).

### 3.5. Toxicity

Treatment-induced toxicities are summarized in Table 4. The most frequently reported toxicity events in this study were vomiting, diarrhea, oral mucositis and neuropathy. Vomiting, oral mucositis and neuropathy were significantly more frequent in the OXA group compared with the GEM group: 51% (11/21), 51% (11/21) and 61% (13/21) versus 10% (5/49, *p* = 0.02), 4% (2/49, *p* = 0.01) and 2% (1/49, *p* = 0.01), respectively. Diarrhea was present in 51% (11/21) of the OXA group patients and 10% (5/49) of the GEM group patients with no significant difference (Table 4). No difference in treatment toxicities was found according to the sarcopenia status (Table 5).

## 4. Discussion

This retrospective study involving patients with resected PDAC followed by adjuvant chemotherapy showed a significant difference in the OS between sarcopenic and non-sarcopenic patients. Sarcopenic patients had a significantly shorter median OS (25 months) compared to non-sarcopenic patients (25 versus 157 months; HR: 6.9; 95% CI: 1.67–28.29) (*p* = 0.01). No statistical differences in DFS were found between the two groups. This highlights the influence of sarcopenia on the OS of patients with initially resectable PDAC.

In our study, 26% of our patients with PDAC were sarcopenic, which is lower than the 35% to 65% rates reported in the literature [7]. This could be explained by the inclusion of only patients with resectable PDAC at an early stage of the disease by definition. In addition, as the Cochin hospital is a tertiary center, a noticeable proportion of PDAC was early detected from the follow-up of intraductal papillary mucinous tumors. The global evaluation of sarcopenia considers not only muscle mass but also strength and physical performance [14,15,16]. In this regard, clinical variables such as BMI, weight loss and food intake allow an effective estimation of the sarcopenic status [17,18]. However, one advantage of CT is that it overcomes some limitations of anthropological measurements, such as being overweight or obese, which can underestimate sarcopenia [19]. Chemotherapy doses are based on the body surface area, and toxicities occur more frequently and more severely when the body surface area does not reflect the nutritional status [19]. By contrast, CT imaging helps identify patients who need specific nutritional support by detecting sarcopenia, therefore allowing them to be less prone to chemotherapy-related adverse events [20]. One current limitation of CT measurements is the time needed for contouring, but artificial intelligence software can automatically make psoas muscle measurements and allows SMI calculation [21,22].

In our study, we found no significant differences in terms of the OS and DFS between patients with resected PDAC who received adjuvant oxaliplatin-based (modified FOLFIRINOX and FOLFOX) chemotherapy and those who received gemcitabine-based (gemcitabine and gemcitabine–capecitabine) chemotherapy. This difference of approximately eight months in DFS between OXA and GEM groups is similar to that reported by Conroy et al. [5]. This emphasizes the representativeness of our study. However, the median OS in the GEM group in our study (157 months) is longer than that in Conroy et al. study (35 months). This difference may be explained by the fact that gemcitabine was the adjuvant therapy used before modified FOLFIRINOX. The patients treated with gemcitabine-based chemotherapy were most likely to have a loss of follow-up, were censored in the Kaplan–Meier analysis and could have live longer than the modified FOLFIRINOX-treated patients.

In our study, we found significant differences in the median OS and DFS between patients with an R0 resection and those with an R1 resection. Similarly, we found significant differences in the median DFS between patients with N0 (66.8 months), N1 (43.4 months) and N2 (27.4 months) status, as reported in other studies [23,24]. In more advanced PDAC, chemotherapy and radiochemotherapy before surgery are used to reduce the risk of R1 resection and improve the OS [25,26,27]. This emphasizes the role of neoadjuvant therapy to induce a tumor response, increase the R0 rate and treat occult micrometastatic disease. The ongoing PANACHE 01 trial should provide some answers regarding the role of neoadjuvant chemotherapy in localized pancreatic adenocarcinoma (ClinicalTrials.gov Identifier: NCT02959879).

Regarding toxicity, we found that modified FOLFIRINOX is more often associated with nausea, neuropathy and oral mucositis than gemcitabine. In our work, there was a 50% incidence of chemotherapy-related adverse events of any grade, which is similar to the toxicity rates reported by Conroy et al. [5]. In addition, the median duration of chemotherapy was significantly lower in the OXA group (4.5 months) vs. the GEM group (6 months). This may be explained by the higher toxicity of FOLFIRINOX, which encourages the discontinuation of the chemotherapy. Contrary to previously reported results [7,8], we found that sarcopenic patients did not have significantly more adverse events than non-sarcopenic ones, which is questionable and possibly due to under-reporting. As gemcitabine generates less toxicity, the oncologist may be more prone to give this chemotherapy to frail patients, which creates an indication bias. 

We found that the median OS of non-sarcopenic patients treated with gemcitabine (157.7 months) was significantly greater than the median OS of sarcopenic patients treated with oxaliplatin-based chemotherapy (14.4 months). However, no significant differences in DFS were found between these two groups of patients. Sarcopenia combined with aggressive oxaliplatin chemotherapy could expose patients to an even lower OS than with gemcitabine. Accordingly, the oncologist must take into consideration the patient’s sarcopenia status before prescribing chemotherapy. 

Our study has several limitations. First, this study was retrospective and observational, which led to measurement bias. For example, performance status and clinical and biological adverse events were not reported for many patients, thus not allowing us to perform an analysis of this category of events. Second, the long study period (2004–2021) covered old guidelines without adjuvant chemotherapy, gemcitabine-based chemotherapy and, more recently, mFOLFIRINOX. This can lead to an indication bias. Third, this study has a small sample size and was conducted in a single institution, which limits the generalizability of our results. Furthermore, other cachexia-related parameters, including food intake or hypoalbuminemia, were not detailed. A comparison between those parameters could have been worthwhile. As well, the patient’s performance status was not reported, which is highly correlated with sarcopenia and could lead to an indication bias. Additionally, a low number of 117 patients was included in our tertiary center; some patients may have been lost. Finally, as seen before, this study was performed in a tertiary center specialized in the detection of early PDAC, increasing a recruitment bias. 

## 5. Conclusions

Among patients with localized PDAC who undergo surgery and are treated with adjuvant chemotherapy, sarcopenia leads to a significantly shorter OS. Prospective studies must be performed to specify its role in overall patient care.

## Figures and Tables

**Figure 1 cancers-14-06179-f001:**
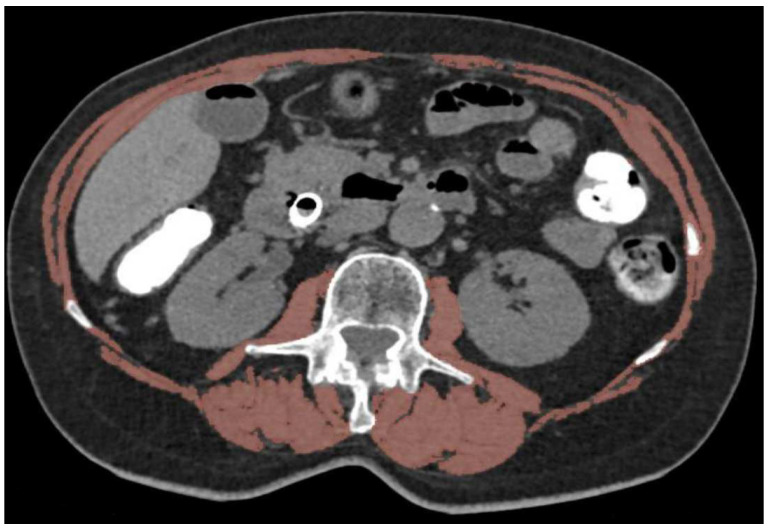
CT image in the axial plane shows segmented muscles in a sarcopenic patient (body mass index = 24 kg/m^2^). Muscle structures forming the cross-section areas used to calculate skeletal muscle index are displayed in orange.

**Figure 2 cancers-14-06179-f002:**
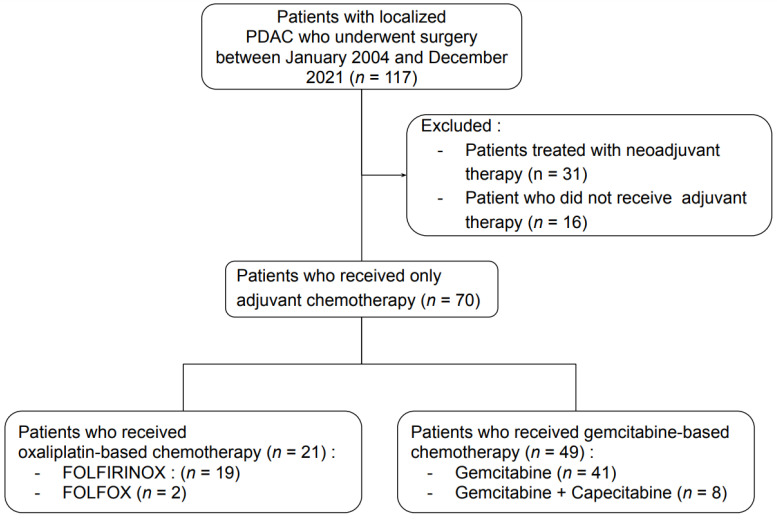
Study flow chart. PDAC indicates pancreatic ductal adenocarcinoma.

**Figure 3 cancers-14-06179-f003:**
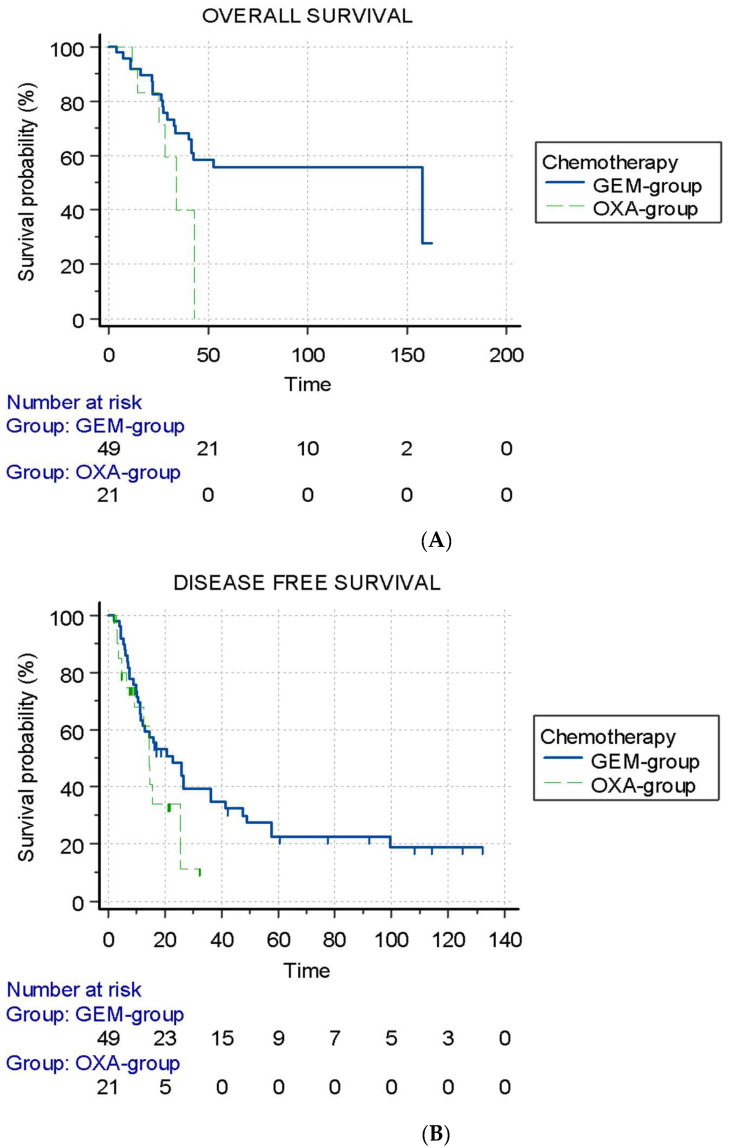
Graphs showing Kaplan–Meier estimates of overall survival (OS) and disease-free survival (DFS) according to the chemotherapy group. (**A**) Graph shows OS in gemcitabine-based(GEM group) and oxaliplatin-based (OXA group) chemotherapy groups. Median OS was 157.7 months in GEM group (IQR: 30 not reached) vs. 34.1 months in OXA group (IQR: 25–45) (*p* = 0.13), yielding hazard ratio of 2.48 (95% CI: 0.77–7.96). (**B**) Graph shows DFS in GEM group and OXA group. Median DFS was 22.7 months in GEM group (IQR: 10.3–57.7) vs. 14.4 months in OXA group (IQR: 7.6–25.5) (*p* = 0.11), yielding hazard ratio of 1.86 (95% CI: 0.87–3.94). Tick marks indicate censored data. Time is expressed in months.

**Figure 4 cancers-14-06179-f004:**
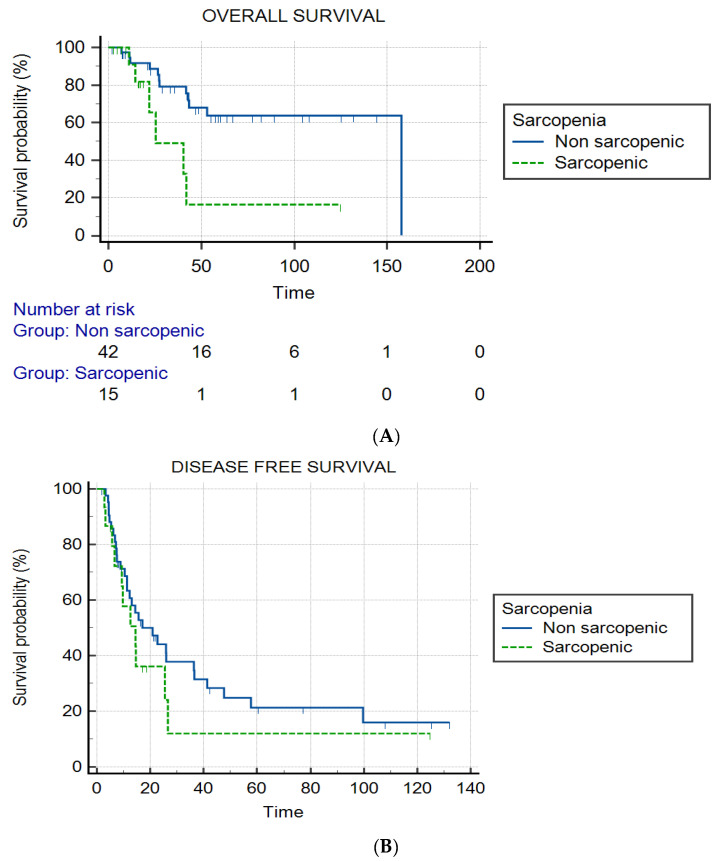
Graphs show Kaplan–Meier estimates of overall survival (OS) and disease-free survival (DFS) according to sarcopenia status. (**A**) Graph shows OS in sarcopenic group and non-sarcopenic group. Median OS was 25.5 months for sarcopenic patients (IQR: 14.5–41.5) vs. 157.7 months for non-sarcopenic patients (IQR: 42.5–157.7) (*p* = 0.01). Hazard ratio was 6.9 (95% CI: 1.67–28.29). (**B**) Graph shows DFS in sarcopenic group and non-sarcopenic group. Median DFS was 20.9 months for non-sarcopenic patients (IQR: 7.4–47.6) vs. 14.6 months for sarcopenic patients (IQR: 6.6–25.9) (*p* = 0.34). Tick marks indicate censored data. Time is expressed in months.

**Figure 5 cancers-14-06179-f005:**
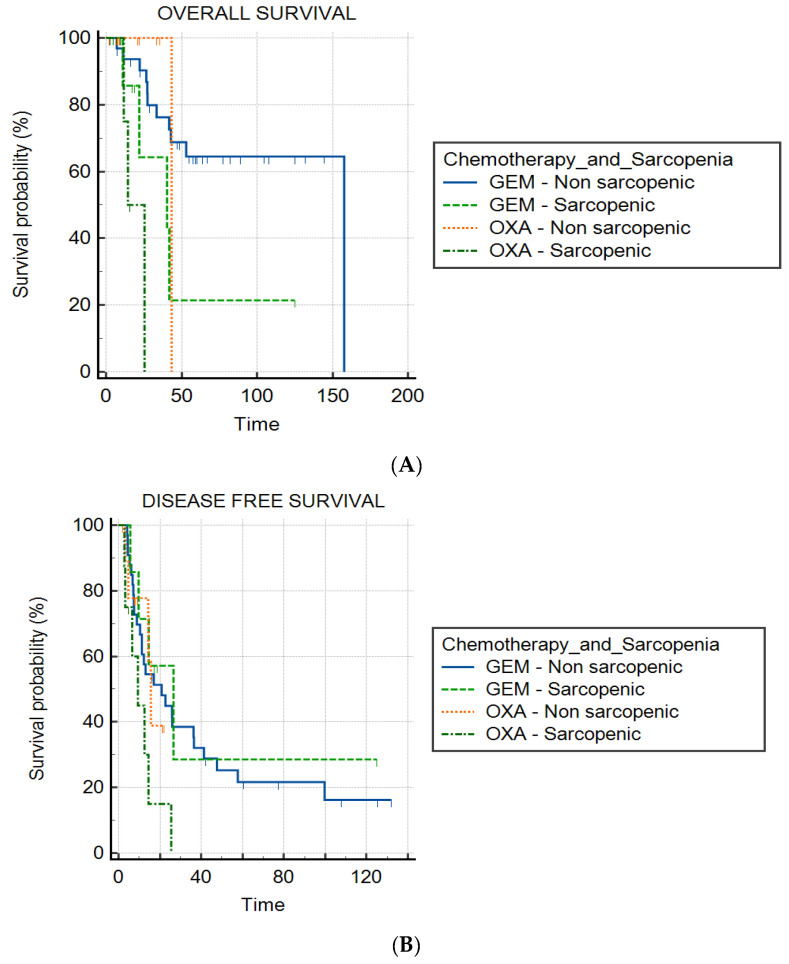
Graphs show Kaplan–Meier estimates of overall survival (OS) and disease-free survival (DFS) according to sarcopenia status and type of chemotherapy. (**A**) Median OS was 157.7 months for the GEM non-sarcopenic group, 43.4 months for the OXA non-sarcopenic group, 40.4 months for the GEM sarcopenic group and 14.4 months for the OXA sarcopenic group (*p* < 0.01). (**B**) Median DFS was 26.6 months in the GEM sarcopenic group, 20.9 months in the GEM non-sarcopenic group, 15.6 months in the OXA non-sarcopenic group and 9.3 months in the OXA sarcopenic group (*p* = 0.07). Tick marks indicate censored data. Time is expressed in months.

**Table 1 cancers-14-06179-t001:** Characteristics of the study population according to the type of adjuvant chemotherapy.

	GEM Group (*n* = 49)	OXA Group (*n* = 21)
Gender		
Male	21 (43)	12 (57)
Female	28 (47)	9 (43)
BMI (kg/m^2^)	25 [18–31]	24 [20–33]
Median age at surgery (year)	67 [45–85]	72 [43–81]
Median CA19-9 at diagnosis (UI/mL)	214 [0–4172]	673 [4–16000]
Biliary stent	22 (44)	7 (33,3)
Sarcopenia	8 (16)	7 (33)
Type of surgery		
Cephalic duodenopancreatectomy	38 (77)	16 (77)
Distal pancreatectomy	11 (23)	5 (23)
Status of surgical margin		
R0	44 (90)	21 (100)
R1	5 (10)	0 (0)
pTNM		
T1	2 (4)	0
T2	12 (24)	12 (57,1)
T3	31 (62)	9 (42,9)
T4	5 (10)	0 (0)
N0	9 (18)	7 (33.3)
N1	35 (70)	7 (33.3)
N2	6 (12)	7 (33.3)
M0	48 (98)	21 (100)
M1	1 (2)	0 (0)
Days between CT and surgery	20 [2–92]	18 [3–98]
Days between surgery and first cycle of chemotherapy	48 [16–414]	55 [21–111]
Duration of chemotherapy (months)	6 [2–6]	4.5 [1–6]

pTNM is defined according to the American Joint Committee on Cancer AJCC 8th edition as: T1 (tumor ≤ 2 cm in the greatest dimension), T2 (tumor between 2 and 4 cm in the greatest dimension), T3 (tumor more than 4 cm in the greatest dimension) and T4 (tumor involves celiac axis or superior mesenteric artery). Nodal status was defined as N0 (no lymph node involvement), N1 (lymph node involvement between 1 and 3) and N2 (lymph node involvement more than 4). M0 is no metastasis and M1 is metastasis. Resection statuses were defined as R0 (no cancer cells within 1 mm of all resection margins) and R1 (cancer cells present within 1 mm of one or more resection margins). All quantitative variables are expressed as medians; numbers in brackets are ranges. Qualitative data are expressed as raw numbers; numbers in parentheses are percentages. GEM group: gemcitabine-based chemotherapy group. OXA group: oxaliplatin-based chemotherapy group.

**Table 2 cancers-14-06179-t002:** Characteristics of the study population according to the sarcopenia status.

	Non-Sarcopenic (*n* = 43)	Sarcopenic (*n* = 15)
Gender		
Male	20 (48)	8 (53)
Female	23 (53)	7 (47)
BMI (kg/m^2^)	24 [18–30]	25 [20–32]
Median age at surgery (years)	65 [43–85]	73 [54–80]
Median CA19-9 at diagnosis (UI/mL)	129 [3–7938]	32 [0–16000]
Biliary stent	17 (40)	7 (47)
Chemotherapy group		
GEM group	32 (74)	8 (53)
OXA group	11 (26)	7 (47)
Type of surgery		
Cephalic duodenopancreatectomy	33 (76)	11 (73)
Distal pancreatectomy	10 (24)	4 (27)
Status of surgical margin		
R0	40 (93)	14 (93)
R1	3 (7)	1 (7)
pTNM		
T1	2 (5)	0 (0)
T2	12 (29)	8 (53)
T3	26 (60)	7 (47)
T4	3 (7)	0 (0)
N0	9 (21)	5 (33)
N1	27 (62)	7 (47)
N2	7 (17)	3 (20)
M0	43 (100)	15 (100)
M1	0 (0)	0 (0)
Days between CT and surgery	20 [2–98]	19 [4–65]
Days between surgery and first cycle of chemotherapy	53.5 [17–414]	51 [31–111]
Duration of chemotherapy (months)	6 [1–6]	6 [2–6]

All quantitative variables are expressed as medians; numbers in brackets are ranges. Qualitative data are expressed as raw numbers; numbers in parentheses are percentages. GEM group: gemcitabine-based chemotherapy group. OXA group: oxaliplatin-based chemotherapy group.

**Table 3 cancers-14-06179-t003:** Log-rank test estimates of disease-free survival and overall survival according to tumor status.

	Tumor Status	Median (Months)	IQR	*p*
Disease-free survival	T1	144	144–144	0.29
	T2	22.7	19.2–48.2	
	T3	13	7.1–27.3	
	T4	10.3	9.2 not reached	
	N0	66.8	15.3 not reached	**0.02**
	N1	43.4	6.5–57.7	
	N2	27.4	10.2–30.4	
	R0	25.5	9–48.6	**<0.01**
	R1	11.3	6.1–13.4	
Overall survival	T1			0.47
	T2	41.9	25 not reached	
	T3	43.4	27–158	
	T4			
	N0		42 not reached	0.10
	N1	158	27.5 not reached	
	N2	34.1	27.5–42	
	R0	158	27.5 not reached	**0.04**
	R1		11 not reached	

Bold indicates significant *p* value.

**Table 4 cancers-14-06179-t004:** Treatment toxicity according to the chemotherapy group.

	Grade	GEM Group	OXA Group	*p*
Nausea, vomiting	1	4 (8)	5 (23)	**0.02**
	2	0 (0)	6 (28)	
	3	1 (2)	0 (0)	
	4	0 (0)	0 (0)	
Diarrhea	1	3 (6)	6 (28)	0.25
	2	2 (4)	5 (23)	
	3	0 (0)	0 (0)	
	4	0 (0)	0 (0)	
Oral mucositis	1	1 (2)	10 (47)	**0.01**
	2	1 (2)	1 (4)	
	3	0 (0)	0 (0)	
	4	0 (0)	0 (0)	
Neuropathy	1	0 (0)	6 (28)	**0.01**
	2	1 (2)	3 (14)	
	3	0 (0)	4 (19)	
	4	0 (0)	0 (0)	

Variables are expressed as raw numbers; numbers in parentheses are percentages. Bold indicates significant *p* value.

**Table 5 cancers-14-06179-t005:** Treatment toxicity according to the sarcopenia status.

	Grade	Non-Sarcopenic	Sarcopenic	*p*
Nausea, vomiting	1	5 (12)	2 (13)	0.40
	2	3 (7)	2 (13)	
	3	1 (2)	0 (0)	
	4	0 (0)	0 (0)	
Diarrhea	1	7 (16)	1 (7)	0.23
	2	2 (5)	3 (20)	
	3	0 (0)	0 (0)	
	4	0 (0)	0 (0)	
Oral mucositis	1	5 (12)	4 (26)	0.72
	2	1 (2)	0 (0)	
	3	0 (0)	0 (0)	
	4	0 (0)	0 (0)	
Neuropathy	1	2 (5)	3 (20)	0.68
	2	1 (2)	2 (13)	
	3	2 (5)	1 (7)	
	4	0 (0)	0 (0)	

Variables are expressed as raw numbers; numbers in parentheses are percentages.

## Data Availability

Not applicable.

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
