# Peer review of "Impact of Sarcopenia on Patients with Localized Pancreatic Ductal Adenocarcinoma Receiving FOLFIRINOX or Gemcitabine as Adjuvant Chemotherapy"

_cancers, 2022, doi:10.3390/cancers14246179_

Round 1

Reviewer 1 Report

In this retrospective cohort study of 70 patients with PDAC treated with upfront resection followed bij adjuvant chemotherpay (gemcitabine based or oxaliplatin based) sarcopenia was associated with worse overall survival.

As sarcopenia (measured with CT segmentation of muscles at the L3 level) is usually not taken into account as casemix correction factor in national audits, this is a sound and important massage.

Having said this I have some major concers about the manuscript including its methodology.

1. I would suggest to only study sarcopenia as a risk factor for overall survival in PDAC taking into account all treated patients, including those who did not receive adjuvant chemotherapy. Outcome data should than be corrected for known prognostic factors in PDAC. The present retrospective study design with a high selection bias and long study period (2004-2021) covering old guidelines without adjuvant chemotherapy, gemcitabine based chemotherapy and more recently mFOLFIRINOX, is clearly not the design to study (survival) differences between the two chemotherapy regimens.

2. As toxiciy profiles are known to be different between gemcitabine based chemotherapy and mFOLFIRINOX, the toxicity paragraph and data in table 4 do not add to the discussion

3. a total of 117 patients were resected in a study period of 17 years. This tertiary reference center should discuss the low numbers (low volume hospital) which could be another bias in the results

4. Sarcopenia was defined at baseline: however this could be as long as 227 days before surgery (table 1). I would suggest to include only CT scans no longer than 6-8 weeks before surgery.

5. the overall survival curve in figure 4. should be corrected for N and R status, since both factors are related to survival (table 3).

6.  I would suggest to also investigate the muscle quality (HU of muscles, myosteatosis) instead of muscle mass at the L3 level.

7. with respect to 6. in the references, I miss  studies on body composition and outcome in pancreatic cancer, for example van Dijk DP et al. HPB 2018, and J Cachexia Sarcopenia Muscle 2017.

Author Response

Thanks for your review.

Afortunately, I don't have the data on the patient who did not receive adjuvant chemotherapy, but it could be interesting of course.

It is our main limitation to be retrospective and with a long study period.

The toxicity part according to chemotherapy, is effectively not necessary. It can be deleted from the article. 

The delay between the diagnostic and the surgery is exact but not interesting. But the delay between the pre operative CT was shorter, I change the table for days between CT and surgery 

I don't have enough data to correct for N and R on the survival curves.

I add some others limitations of the study.

Reviewer 2 Report

This manuscript reports a retrospective cohort of 70 patients who underwent surgery for pancreatic ductal adenocarcinoma (PDAC), and subsequently received adjuvant chemotherapy, over a 3 year period. The relationship between sarcopenia, as measured on a baseline CT scan and overall survival, disease specific survival and toxicity is examined.

The abstract provides a clear and helpful summary of the work undertaken and conclusions drawn.

The introduction gives helpful background information regarding the disease and role of adjuvant chemotherapy. Some of the historical background to adjuvant chemotherapy is missing- the first treatment of proven benefit was actually 5FU with folinic acid (ESPAC1 trial). The statement that mFOLFIRINOX currently remains the standard of care in his setting perhaps needs to be qualified by 'in patients fit enough to receive it'- which in my experience is by no means all. Helpful background is also provided regarding the importance of sarcopenia in PDAC.

The methods section clearly presents inclusion and exclusion criteria for the study. It is stated that 'patients were assigned to gemcitabine based or oxaliplatin based chemotherapy groups'. It would be helpful to include some comment as to what factors led to selection of patients for each treatment. Given the timeframe in which the study was completed presumably some of this was driven by date of entry as mFOLFINIOX is a relatively more recent treatment in this field, but presumably patient fitness/performance status will also have played a role more recently.

A clear definition of sarcopenia in terms of skeletal muscle index is given, along with a description of how this is calculated. However, it is not clear at what time point the baseline CT was undertaken- was this pre- or post-operative?

It is stated that tumour evaluation was performed every 4 cycles of chemotherapy, and after chemotherapy every 3 months. It is not clear what this evaluation comprised- was it clinical, biochemical, radiological?

I do have some concerns about the appropriateness of the statistical analysis undertaken, in particular regarding the comparison undertaken between GEM- and OXA- based chemotherapy groups. Given the extensive confounding factors leading to choice of treatment already alluded to, I'm not sure this is helpful or worthwhile and feel it would probably be better to stick with the main goal of the paper, comparing outcomes in patients with or without sarcopenia. Type of chemotherapy received could perhaps be considered as a part of a multi-variate analysis?

The results are in general very clearly reported, although, as above, I am not convinced that it is appropriate to present the GEM- vs OXA- comparison in this way. Figure 2 is clear and particularly helpful. As above, I think it would be helpful to consider a multivariate analysis of relevant factors for the whole group, including sarcopenia, type of chemotherapy received, performance status, T/N stage and R0/1 status which are all known prognostic factors. This would demonstrate whether knowing about sarcopenia holds any independent prognostic significance.

In the discussion, the conclusions drawn about relationship between sarcopenia and the primary outcome are clear and reasonable. Regarding DFS, it would strictly be more accurate to say that no statistically significant difference was found. As above, I am really not sure that relevant conclusions can be drawn from the GEM- vs OXA- comparison, neither that the comparison of non-sarcopenic GEM patients with sarcopenic OXA- patients adds anything. 

Limitations of the study are appropriately recognised, in particular its retrospective nature. In relation to this it would be worth specifically commenting that this will have affected toxicity reporting. I would also suggest that the absence of any data on performance status is a major limitation- especially since this may well be highly correlated with sarcopenia and may be a much easier to measure variable.

Author Response

I add some limitations and reference.

CT scan was preoperative.

As the number of patients was not sufficient to perform a multivariate analysis, i can not included one in the paper. but I agree with it is preferable.

Effectively, comparaison between chemotherapy group may not stick to the main goal. 

Round 2

Reviewer 1 Report

I'm happy with this latest version.

The authors addressed my concerns sufficiently by adapting the discussion session.

Author Response

We thank you for your comments and your expertise

Reviewer 2 Report

A number of my previous comments have not really been addressed, in particular concerns about appropriate of comparisons between groups. 

The author's responses are very brief and do not adequately respond to my comments

Author Response

Dear reviewers,

We would like first to thank the reviewer for their expertise on our manuscript and for their propositions of improvement.

We paid particular attention to all reviewers’ comments. Some references have been added in the revised manuscript.

The text was modified to answer all comments from the editor and the reviewers. Please find attached a point-by-point response to reviewer.

Yours sincerely
